# Exploring Biodiversity through the Lens of *Knautia arvensis* Pollinators: *Knautia* Pollinator Walks as a Monitoring Method

**DOI:** 10.3390/insects15080563

**Published:** 2024-07-25

**Authors:** Markus Franzén, Magnus Stenmark

**Affiliations:** 1Department of Physics, Chemistry and Biology (IFM), Linköping University, SE-581 83 Linköping, Sweden; 2Center for Ecology and Evolution in Microbial Model Systems, EEMiS, Department of Biology and Environmental Science, Linnaeus University, SE-391 82 Kalmar, Sweden; 3Calluna AB, Linköpings Slott, SE-582 28 Linköping, Sweden; magnus.stenmark@calluna.se

**Keywords:** agricultural systems, biodiversity indicators, conservation strategies, *Knautia arvensis*, monitoring methods, pollinator communities, pollinators, Russia, solitary bees, Sweden

## Abstract

**Simple Summary:**

Monitoring pollinator populations is crucial for understanding biodiversity trends and ensuring the health of ecosystems, especially in agricultural landscapes. This study introduces the “*Knautia* Pollinator Walk” as a new method for tracking pollinator diversity and abundance. By observing pollinators visiting the inflorescence of *Knautia arvensis*, we found significant correlations between pollinators and land use, and there were significant differences in pollinator communities between regions. Our findings highlight the importance of habitat type in influencing pollinator populations, offering a valuable tool for conservation efforts.

**Abstract:**

Declining populations of native pollinators, especially wild bees, underline the urgent need for effective monitoring within agricultural ecosystems. This study aims to (i) establish the ‘*Knautia* Pollinator Walk’ as an innovative pollinator monitoring method, (ii) examine the link between pollinator richness/density and land cover, and (iii) assess if specialist solitary bees indicate pollinator abundance and morphogroup richness. The approach involves surveying 500 *Knautia arvensis* inflorescences per site thrice per season. Observations of 11,567 pollinators across 203 taxa showed significant correlations between pollinator diversity and land use. Pollinator populations fluctuated with land cover type, increasing in open areas but decreasing or stabilising in forested and shrubby regions. Noteworthy differences in pollinator types were seen between Russia (solitary bees, small Diptera, Lepidoptera) and Sweden (bumblebees, beetles, furry Diptera). The “*Knautia* Pollinator Walk” shows promising signs of being an effective tool for monitoring spatiotemporal biodiversity trends. The method offers a scalable approach to pollinator monitoring, which is essential for developing conservation strategies and supporting pollinator populations.

## 1. Introduction

Biodiversity, critical for human well-being, is undergoing a global decline at an alarming rate, characterised by increasing species extinctions and ecosystem degradation [1]. This escalating crisis has catalysed political commitments to mitigate its impacts [2,3]. A major challenge in assessing population trends among insects, including key pollinators, lies in their significant population fluctuations [4,5,6,7]. These fluctuations have been attributed to meteorological conditions, landscape-level factors such as agricultural land cover, and a generation time spanning over multiple years [4,5,8,9]. Recently, a notable shift in the populations of butterflies and moths has been documented, yet pollinators remain under-researched in this context [10,11,12,13,14].

Pollinators, indispensable for most flowering plants and many crops, are at the forefront of global conservation efforts [15,16,17,18]. Western Europe’s landscape has significantly been transformed by high-intensity land use, which presents a stark contrast to parts of Eastern Europe, where lower land-use intensity probably supports larger populations and more diverse pollinator communities [19,20]. Specialised solitary bees, sensitive to environmental changes, are considered key indicators of ecosystem health [21]. Despite their ecological importance, gaps in knowledge and standardised monitoring methods for pollinators persist, hindering effective conservation, and data about long-term trends of pollinator communities are scarce [18,19,22,23]. Observations of pollinators on focal plants have emerged as a promising and standardised method for pollinator monitoring [24].

Monitoring mechanisms for wild bee populations are essential to identify pollinator hotspots and assess biodiversity trends and ecosystem health. Pollen specialist solitary bees in generalist pollination systems might be promising biodiversity indicators [25]. The gynodioecious herb *Knautia arvensis* (Dipsacaceae), frequently visited by diverse pollinators, is an example of a generalist pollination system, where species like *Andrena hattorfiana* and *Dasypoda suripes* are declining, indicating broader ecological challenges [26,27,28,29]. Understanding pollination dynamics, particularly in systems with generalist and specialist pollinators, is crucial to understanding biodiversity trends and the community composition of pollinators [30,31].

*Knautia arvensis*, commonly known as field scabious, is an excellent plant for monitoring pollinators due to its widespread distribution and ease of recognition. This perennial herbaceous species attracts a diverse array of pollinators, including bees, butterflies, and hoverflies, making it a valuable indicator of pollinator activity [32,33,34,35]. Its adaptability allows it to thrive in both wild and garden settings, and it can be easily sown or favoured through targeted management practices [36]. These characteristics make *K. arvensis* a practical and effective choice for pollinator monitoring programs, contributing significantly to biodiversity assessments and conservation strategies.

Consistent, standardised monitoring of flower-visiting insect populations is crucial due to their dynamic nature and essential role in pollination and ecosystem health, with current programs highlighting both progress and challenges [37,38,39]. This study seeks to confront the outlined challenges by introducing a systematic, efficient, cost-effective approach to monitoring pollinators—the *Knautia* pollinator walk. Utilising data from the *Knautia* pollinator walk, our primary objectives were (i) to introduce and establish the ‘*Knautia* Pollinator Walk’ as a novel method for pollinator monitoring, (ii) to ascertain the correlation between morphogroup richness and density of pollinators and specific land cover types, and (iii) to explore if specialist solitary bees are indicators of high pollinator densities.

## 2. Material and Methods

### 2.1. Study Area and the ‘Knautia Pollinator Walk’

Our study spanned 85 sites, visited thrice each season, across Sweden (77 sites) and Russia (8 sites), from 2004 to 2017, extending from latitude 54° to 63° and longitude 13° to 49° (Figure 1). Eligibility criteria for site selection required a minimum of 500 inflorescences and a separation of at least 500 m from the nearest conspecifics. All sites were grassland sites either abandoned, grazed, or mowed. In instances where the requisite number of 500 *K. arvensis* inflorescences is not attainable at a single survey site due to livestock grazing, natural grazing, mowing, or habitat destruction, the existing inflorescences were repeatedly surveyed. This repetition was continued until the equivalent data for 500 unique inflorescences were collected. Our primary focus was on the peak flowering season of *K. arvensis.* Each site, measuring 0.25 to 1 hectare, primarily in Sweden and Western Russia (Figure 1), underwent three methodological *Knautia* pollinator walks within a year. We assessed 500 *Knautia arvensis* flowers per visit in landscapes comprising mixed native vegetation, including farmlands and orchards [40]. When fewer than 500 inflorescences were present at a site, the existing inflorescences were sampled multiple times until the equivalent of 500 unique inflorescences was reached. On a few occasions, this was necessary due to cutting or management practices. Pollinator walks, conducted exclusively by the authors, were employed for surveying flower visitors at each plant population, performed thrice (early-, mid-, and late-season) at regular intervals from June 20 to August 7 (Figure 2). One pollinator walk was normally conducted during 20–35 min due to site-specific factors. Visitors were categorised into ten major groups (Figure 2), modified from Larsson [25], with species-level identification achieved for most flower visitors at 77 sites. Sampling was temporarily halted for netting or photographing necessary for identification. When precise identification was not feasible (e.g., due to flying away before being documented), visitors were assigned to genus, family, or order and later categorised into any of the ten pre-defined pollinator groups. The study highlights a stark contrast in land-use intensity. Swedish sites typify regions with modern, intensive agricultural and forestry practices. In contrast, the Russian sites represent areas with significantly lower land-use intensity, free from contemporary farming or forestry activities. Fieldwork was conducted exclusively under favourable weather conditions, specifically on days with clear skies, temperatures between 17 and 30 °C, and wind speeds below six m/s.

### 2.2. The Studied Plant Species K. arvensis

*Knautia arvensis*, commonly known as field scabious, is a perennial herbaceous plant in the family Dipsacaceae. It is widely distributed across Europe and Asia, thriving in habitats such as meadows, grasslands, and open woodlands. *Knautia arvensis* typically grows to 30–70 cm. It features a basal rosette of leaves and branching stems with opposite, pinnately lobed leaves. The plant is well-known for its lilac to pale blue flowers, composed of dense, rounded heads of tiny florets, each with four petals. These inflorescences are approximately 2–4 cm in diameter and bloom from June to September [41]. *Knautia arvensis* plays a significant role in the ecosystem as a nectar source for many pollinators, including bees, butterflies, and hoverflies [32,33,34,35]. Its long flowering period makes it a valuable resource for these insects [42]. *Knautia arvensis* is adaptable to various soil types but prefers well-drained, calcareous soils. It can grow in sunny and partially shaded locations, indicating its versatility in different environmental conditions [36]. Given its importance as a pollinator, *Knautia arvensis* is often included in wildflower mixes for habitat restoration and biodiversity conservation projects. It is considered an indicator species for certain types of grasslands and is used in ecological studies to assess habitat quality [43].

### 2.3. Studied Pollinator Groups

Our study focused on ten key pollinator morphogroups with distinct ecological roles and significance. Firstly, *Andrena hattorfiana*, a pollen-specialist solitary bee, is primarily reliant on *K. arvensis* and is threatened in parts of Europe due to habitat changes, including agricultural intensification [27,35,44,45]. Secondly, *Dasypoda suripes*, a steppe species previously found in Eastern Skåne and Öland, Sweden, specialised in *Knautia arvensis* and is now likely extinct in the Nordic region [28,46]. Thirdly, *Apis mellifera* (Honeybee), a significant contributor to flower visits, is essential for extracting nectar from *K. arvensis* but potentially competitive with other pollinators. The fourth group encompasses “other solitary bees”, including mining, leaf-cutter, and mason bees, vital for pollinating wildflowers and crops. Fifth, bumblebees (*Bombus* spp.), large social insects, are effective pollinators due to their size, behaviour, and adaptability to various weather conditions. The sixth group, Coleoptera (beetles), includes scarab, flower beetles, weevils, etc., feeding on nectar and pollen. Seventh, Lepidoptera (butterflies and moths), recognised for their colourful wings, includes species like swallowtails. Eighth, the “furry Diptera”, consisting of hairy hoverflies (Syrphidae), robber flies (Asilidae), bee flies (Bombyliidae), and others that often mimic bees and wasps that feed on nectar and pollen. Ninth, “non-furry Diptera”, which includes many hoverflies, tachinid flies, fruit flies, and mosquitoes, is known for visiting flowers for nectar and pollen. Lastly, the tenth group, “other arthropods”, comprises diverse pollinators like spiders, ants, crustaceans, centipedes, millipedes, springtails, and true bugs. Each group was defined for its unique contribution to the pollination dynamics within the ecosystems surrounding *K. arvensis* [25]. In future monitoring efforts, it is feasible to identify most pollinators to the species level and later categorise them into our ten groups of interest (Appendix A).

### 2.4. Datasets

We divided the data into three sets to accommodate the varying capabilities for species identification and the availability of land cover data across regions. This partitioning was necessary to accommodate the different capabilities for species identification and the differing availability of land cover data across regions. Each dataset is tailored for distinct analytical purposes, enabling a comprehensive exploration of various tests and hypotheses within the constraints of our available data. Dataset 1 allows for an analysis of the species richness of flower visitors as all flower visitors were identified at the species level. Dataset 2: Abundance and Taxonomic Groupings within Sweden where land cover data were available. It includes the identification of the ten pollinator groups’ land cover data, making pollinator group richness with land cover analyses possible, with 77 sites. Dataset 3 includes a comparative analysis across Sweden and Russia and comprises data from 77 sites in dataset 2 and 8 additional sites from Russia, where all flower visitors have been assigned to any of the ten pollinator groups to explore differences between sites in pollinator group frequency.

### 2.5. Land Cover Data Acquisition

Land cover data pertinent to our study were meticulously extracted from the Swedish Land Cover Database [47]. The variables extracted included the proportion of vegetative other open land, forest cover, shrub cover, and ground moisture index. These landscape metrics were derived for a total of 77 sites. For each site, data collection encompassed a surrounding buffer area with a radius of 100 m, effectively covering an area of approximately 314 square meters per site. This approach ensured a comprehensive landscape-level analysis, providing detailed insights into the land cover characteristics proximal to each site under investigation.

### 2.6. Statistical Analyses

Our study incorporated a range of ecological predictors, including the proportion of vegetative other open land, forest cover, shrub cover, ground moisture index, and latitude. Latitude was particularly emphasised, given its frequent influence on biological communities. To comprehensively assess the impact of these predictors, we constructed three separate linear models (LMs) using the lm function in R as the data were normally distributed. Each model targeted a specific response variable: species richness, richness of taxonomic groups, and pollinator density. Including quadratic terms for all predictor variables allowed us to explore potential non-linear relationships. Model selection was rigorously conducted using Akaike Information Criteria (AIC), as Akaike [48] proposed. The model with the lowest AIC was selected in each case, ensuring optimal model parsimony and fit. Notably, the data followed a normal distribution, justifying using a Gaussian distribution within our LMs. This normality underscores the appropriateness of our selected statistical approach. Our models thus tested the association between our response variables (species richness, taxonomic group richness, and pollinator density) and our suite of continuous predictor variables (ground moisture index, tree and shrub cover, and vegetative other open land). ANOVA was conducted to ascertain variations in pollinator density and group richness between the two countries and to explore whether the specialist bees are good indicators of high pollinator density. The ten pollinator groups were compared between the two countries and across the ten groups using ANOVA and post hoc tests. For indicators of species richness, the presence or absence of specialist bees was related to pollinator richness and density using ANOVA.

To evaluate if three visits per site and season detect most pollinator groups, we conducted a species accumulation curve analysis using the specaccum function from the ‘vegan’ package, using a ‘random’ method. This method involves randomly reordering the sampling units (in this case, sites) and calculating the cumulative number of distinct taxonomic groups (species richness) observed with each additional unit. This process is repeated multiple times, and the average species richness for each level of sampling effort (number of sites visited) is recorded. Data were analysed using R version 4.3.0 [49].

## 3. Results

### 3.1. Pollinator Diversity and Prevalence across Sites

Within the 77 sites where flower visitors were surveyed to species level, 7470 individuals were identified, encompassing 203 taxa (Appendix A). The most common species was the beetle *Leptura melanura,* followed by the non-furry Diptera *Phaonia basalis*. The species richness varied considerably, ranging from five species in Vickleby to 43 species in Råshult and Åryd, with an average of 27 species per site during three visits. In a broader analysis of 85 sites, which included 11,567 pollinator encounters, we observed a distinct dominance of four pollinator groups. The most prevalent were Lepidoptera with 2319 visitations, closely followed by Non-furry Diptera with 2299, Bumblebees with 2284, and Coleoptera with 2242 visitations.

Conversely, *Dasypoda suripes*, a specialist bee on *K. arvensis*, was notably less common, recorded only six times, underscoring its rarity in the regions surveyed. The number of flower visits also varied significantly across sites, with a low of 62 visits recorded in Vickleby and a high of 744 in Skansåsa and Bohult, highlighting substantial site-to-site variability in pollinator activity. Three visits detected most taxonomic groups (Appendix A).

### 3.2. Flower Visitor Species, Taxonomic Group Richness, and Density of Land Cover

Our analysis revealed distinct patterns in pollinators’ species richness and density about land cover types. Species richness declined with increasing shrub cover, showing a plateau at higher levels of coverage. Interestingly, richness tended to rise with an increase in the extent of vegetative other open land, suggesting a preference for these habitats among pollinators (Figure 3, Table 1). Regarding taxonomic group richness, we observed a decrease in conjunction with rising forest cover; however, this trend reversed, showing an increase in richness at higher forest coverage levels. Taxonomic group richness was also positively associated with vegetative other open land (Figure 3A, Table 1. Pollinator density was negatively impacted by increasing ground shrub cover, with a notable decline observed as the cover became denser. A similar decreasing trend was evident with rising ground moisture, indicating that drier conditions may be more conducive to higher pollinator densities (Figure 3C, Table 1).

### 3.3. Contrasting Pollinator Frequencies between Sweden and Russia

Our comparative analysis between Sweden and Russia revealed striking differences in pollinator frequencies (Figure 4). In Russia, Lepidoptera was the most prevalent group, constituting 38.2% of pollinator visits, while in Sweden, they represented only 18.8%. Conversely, Bumblebees showed a higher frequency in Sweden (21.0%) than 1.2% in Russia. Similarly, Coleoptera and Non-furry Diptera were more common in Sweden, at 20.1% and 20.9%, respectively, compared to 8.56% and 5.35% in Russia. Notably, the specialised *Dasypoda suripes* were observed in Russia but absent in the Swedish sites. Other solitary bees also showed a marked discrepancy, with a significant presence in Russia (26.3%) against a modest 3.29% in Sweden.

Additionally, *Andrena hattorfiana*, although infrequent in both regions, was more commonly found in Russia at 0.12% compared to 0.02% in Sweden. *Apis mellifera* (honeybee) and Furry Diptera frequencies were also more frequent in Russia than in Sweden, with honeybees making up 2.5% of the pollinator frequency in Russia against 0.4% in Sweden, and Furry Diptera at 1.9% in Russia compared to 10.1% in Sweden. Other arthropods were relatively similar across both regions, with 2.9% in Russia and 3.1% in Sweden. Our comparative analyses between Sweden and Russia, examining pollinator density and taxonomic group richness, yielded no significant differences between the two regions (Appendix A). Furthermore, neither specialist bee species indicated richer or denser pollinator sites (Appendix A).

## 4. Discussion

Our data suggest that *Knautia* pollinator walks are essential in understanding pollinator communities. The method of registering every flower visitor on 500 inflorescences of *K. arvensis* at three distinct visits during its flowering season forms a baseline that can be repeated to identify biodiversity trends. Furthermore, the results of 47 pollinator walk sites have yielded significant insights into the mechanisms of biodiversity indicators and regional variations in pollinator taxonomic richness. This study aligns with recent research emphasising the influence of landscape characteristics on pollinator diversity [17]. We observed that pollinator density is inversely correlated with ground moisture levels, a finding that echoes the results of similar studies [50]. Additionally, our results indicate an increase in species richness with a reduction in shrub and tree cover and a lower proportion of open land that is not arable fields, consistent with patterns noted in other research [51]. Notably, the composition of flower visitors exhibited marked differences between landscapes of high and low land-use intensity, supporting the findings of previous studies that highlight the impact of land-use intensity on pollinator communities [52,53,54].

Furthermore, our data revealed a variation in the abundance of pollinator groups between Sweden and Russia, suggesting geographical differences in pollinator assemblages, as discussed by [55]. The potential of this method to track pollinator populations over time is promising, particularly for exploring spatiotemporal patterns. Such longitudinal studies are crucial for understanding pollinator communities’ dynamics, as Carvalheiro et al. [56] emphasised. Overall, this method presents a robust framework for monitoring pollinator populations and can significantly contribute to our understanding of their ecological dynamics and conservation needs.

### 4.1. The Importance of an Easy Method to Monitor Groups, Differences, and Changes

While identifying all pollinator species poses challenges, developing an illustrated app and online reporting system will significantly enhance our ability to identify pollinators accurately for species or at least at the genus level. This feat would be unique in its scope and utility. This approach is exceptionally viable given the widespread distribution of *K. arvensis*, which allows for monitoring from the Mediterranean up to the boreal region. Monitoring pollinators is crucial for understanding their diversity, abundance, and distribution and assessing the impacts of environmental changes on these populations [17]. The simplicity of *Knautia* pollinator walks enables researchers and conservationists to gather data on pollinator populations efficiently. These data are indispensable for formulating informed conservation and management strategies. Standard methods for monitoring pollinators include visual searches, pan trapping, and transect sampling.

Each method has its merits and limitations. Visual searches, though accurate, demand significant time and expertise [57]. Pan trapping, on the other hand, is more straightforward but may only capture some species effectively [58]. Transect sampling balances ease and accuracy but may not match the precision of visual searches [59]. Habitat loss, pesticide use, and climate change profoundly affect pollinator assemblages [60,61,62]. An efficient monitoring method helps detect these changes, providing critical information for conservation efforts. For instance, identifying a decline in a particular pollinator species can direct conservation actions toward protecting or restoring its habitat [16,63,64].

### 4.2. Habitat Transformations and Biodiversity Indicators

The loss of specific pollinator species can significantly affect the dependent plants and the broader ecosystem. For instance, the disappearance of a critical pollinator may lead to a decline in certain plant species, triggering cascading effects on the ecosystem. Additionally, alterations in pollinator assemblages can affect the distribution and abundance of various plant species, as Ollerton, Winfree, and Tarrant [15] noted. Monitoring these pollinator assemblages is vital to detect and respond to such ecological changes. An efficient monitoring method aids in identifying shifts in pollinator assemblages, thus providing essential data for conservation strategies like habitat restoration, pesticide reduction, and adaptation to climate change. This approach is crucial for mitigating the impacts of habitat loss, pesticide use, and climate change on ecosystems and crops. Specialist bees, which rely heavily on specific plants or habitats, are key biodiversity indicators. Their sensitivity to environmental changes makes them effective in monitoring ecosystem health and diversity. For example, population changes in specialist bees can indicate habitat alteration or recovery [17].

Furthermore, variations in the abundance or diversity of these bees offer insights into the overall ecosystem health and interspecies interactions [65]. However, identifying universal indicators of biodiversity can be challenging. While certain species like the Zygaenidae moth family (as discussed by [66]) can be informative, comprehensive studies on pollinator data from *Knautia* and other sources are needed to ascertain suitable biodiversity indicators. We found no evidence that specialist solitary bees are indicators of pollinator density or richness. Instead, the density of *K. arvensis* is a good indicator of high species richness [43], highlighting the importance of the plant in the ecosystem.

### 4.3. The Relationship between Land Use and Pollinator Communities Is Complex and Multifaceted

Disturbed landscapes often lead to disrupted pollinator communities, with some species becoming superabundant due to changes in the availability of resources. For instance, Knautia arvensis can attract many pollinators in landscapes with few flowers, becoming a crucial resource in otherwise impoverished environments. The dynamics differ significantly between high-intensity and low-intensity landscapes. In low-intensity landscapes with natural populations, there is often high competition, predation, and specialisation among pollinators.

Conversely, in these landscapes with abundant floral resources, pollinators may not compete intensely for *Knautia* due to the availability of better options, as indicated by studies on pollinator foraging behaviour [25,67,68]. The impact of land-use intensity on pollinator abundance, richness, and diversity cannot be overstated. High-intensity land use, characterised by urbanisation and intensive agriculture, often leads to habitat destruction and fragmentation, negatively impacting pollinator populations [69].

In contrast, low-intensity land use, with its preservation of natural habitats and creation of corridors, supports pollinator diversity [17]. Additionally, the type and management of vegetation, including the use of pesticides, play a significant role in shaping pollinator communities. While targeting pests, pesticides can also adversely affect pollinators by reducing the availability of flowers and nesting sites [70]. Land use also influences the dominance of certain taxonomic and functional pollinator groups. For example, Coleoptera dominates in forest areas rich in shrubland, which provides suitable habitats [71]. Bumblebees and furry dipterans, effective pollinators, are often found in farmland and forest edges, where they find both nesting and foraging habitats.

Furthermore, the importance of pollination varies among groups. Larsson [25] highlighted the significance of bumblebees in pollinating Knautia arvensis, given their ability to transfer large amounts of pollen across all sexual stages of the plant. In contrast, despite their ecological value, specialist bees are often less critical as pollinators due to their specific pollen removal behaviours and avoidance of certain floral stages. In conclusion, understanding the nuanced relationship between land use, site characteristics, and pollinator assemblages is crucial for effective conservation and management strategies. Acknowledging different pollinator groups’ varied roles and contributions in these ecosystems is critical to developing targeted and impactful conservation measures.

### 4.4. The Imperative of Longitudinal Monitoring of Flower-Visiting Insect Populations

The necessity for consistent, standardised monitoring of insect populations, particularly flower-visiting species, is paramount due to their dynamic nature and critical role in pollination and ecosystem health. The lack of systematic surveillance programs, as highlighted by the severe fluctuations in insect populations noted in various studies, including research on post-drought declines [72,73], indicates a significant oversight. High-quality time series data are essential for documenting these fluctuations and discerning their causes and possible mitigation strategies. The Flower–Insect Timed Counts (FIT Count) protocol and the UK Pollinator Monitoring Scheme (PoMS) represent significant strides in addressing this gap [38,39]. However, PoMS faces challenges, including its limited scope in monitoring rare species, resource-intensive nature, and complexities in data integration [37].

Similarly, The National Inventory of Landscapes in Sweden (NILS) represents a significant effort towards establishing a multiscale biodiversity monitoring system, mainly surveying key pollinator groups such as butterflies and bumblebees [74]. Despite this monitoring system’s comprehensive scope and robust design, the output, in terms of scientific findings and practical applications, has been surprisingly low [75]. Focusing on specific plants like *Knautia arvensis*, a key nectar source for many pollinators, is critical. This approach indicates pollinator community health and offers a more focused and directly comparable monitoring method involving a manageable number of flower visitors for identification.

## 5. Conclusions

Monitoring programs are indispensable for understanding biodiversity trends and aiding conservation efforts. However, a significant limitation of current methodologies lies in the imprecise detection of pollinator fauna due to the need for effective pollinator monitoring programs. The *Knautia* pollinator walks method distinguishes itself by its potential to accurately identify the number and species of pollinators, thereby characterising precise pollinator assemblages. Three *Knautia* pollinator walks per site detected most taxonomic groups, representing a reasonable and realistic frequency for robust data collection. It closely approaches the asymptote of taxonomic group richness, indicating its effectiveness in capturing the breadth of biodiversity present. The precision of *Knautia* pollinator walks renders it a reliable tool for detecting trends at local, regional, and national levels. Characterised by its broad applicability, the method is suitable for a range of environments, from natural ecosystems to managed landscapes such as gardens and parks. It is also adaptable to most European Union countries and a diverse array of habitats. Such an approach is pivotal in capturing current trends in pollinator populations, thereby informing and shaping conservation efforts to support ecological sustainability and food security. We propose that *Knautia* pollinator walks be integrated as a recommended component of national monitoring programmes.

## Figures and Tables

**Figure 1 insects-15-00563-f001:**
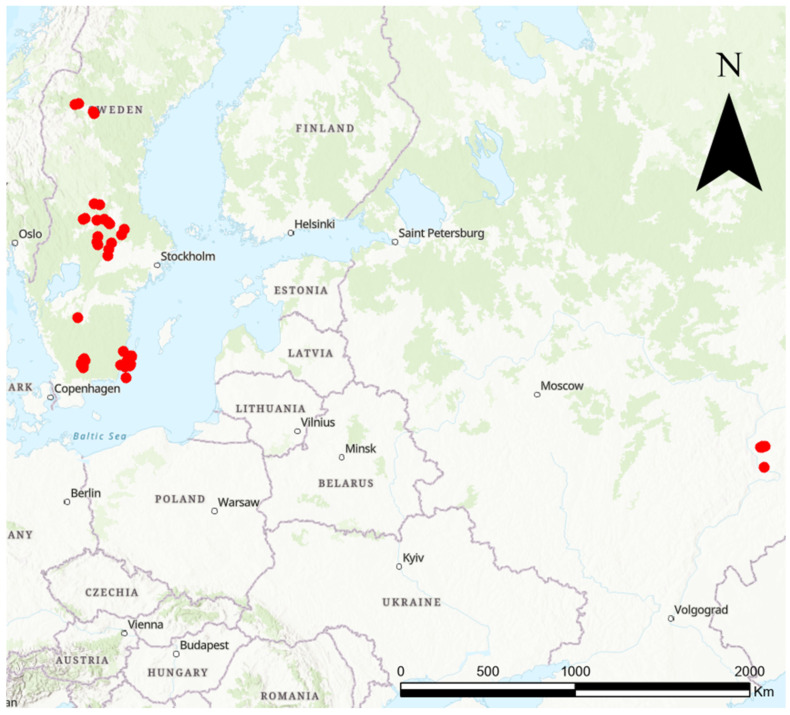
Map of the study area with the 85 studied sites denoted by red dots.

**Figure 2 insects-15-00563-f002:**
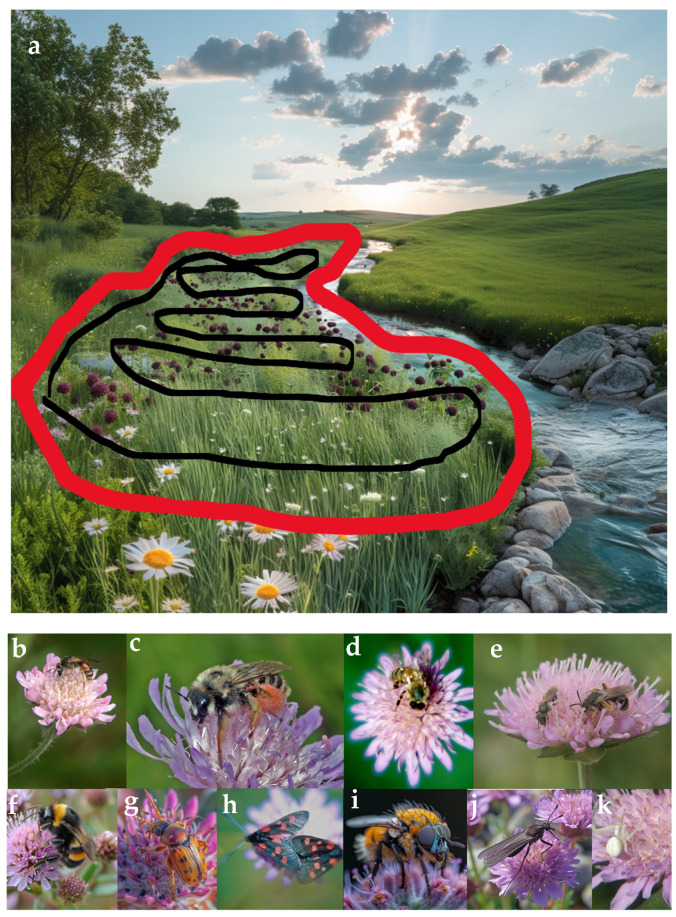
Overview of pollinator monitoring and species observed in the study. (**a**) The *Knautia* 500 Pollinator Walk. This figure illustrates the methodological framework for assessing pollinator diversity and activity around *Knautia* populations. It features the path (black line) within a specified area (outlined in red), showcasing the pollinator walk. (**b**) *Andrena hattorfiana*, a pollen-specialist solitary bee, faces threats in certain European regions and predominantly depends on *K. arvensis* for survival—photo by Magnus Stenmark. (**c**) *Dasypoda suripes*, a steppe species specialised in *K. arvensis*, is now considered likely extinct in the Nordic region—photo by Magnus Stenmark. (**d**) *Apis mellifera* (Honeybee) is crucial for flower pollination through nectar extraction yet poses competition to other pollinators—photo by Magnus Stenmark. (**e**) Other Solitary Bees, including sweat, mining, leaf-cutter, and mason bees, are essential for pollinating wildflowers and crops. (**f**) *Bombus* spp. (bumblebees), noted for their effective pollination capabilities, which are attributed to their size, behaviour, and versatility. (**g**) Coleoptera (beetles), including various nectar and pollen-feeding species such as scarabs, flower beetles, weevils, and fireflies. (**h**) Lepidoptera (butterflies and moths), distinguished by their vividly coloured wings and including species like swallowtails. (**i**) Furry Diptera, comprising bee and wasp mimics such as hairy hoverflies, robber flies, and bee flies. (**j**) Non-furry Diptera, covering a range of flower-visiting flies, including hoverflies, tachinid flies, fruit flies, and mosquitoes. (**k**) Other Arthropods featuring a diverse array of pollinators, including spiders, ants, crustaceans, centipedes, millipedes, springtails, and true bugs. Photos/illustrations Markus Franzén.

**Figure 3 insects-15-00563-f003:**
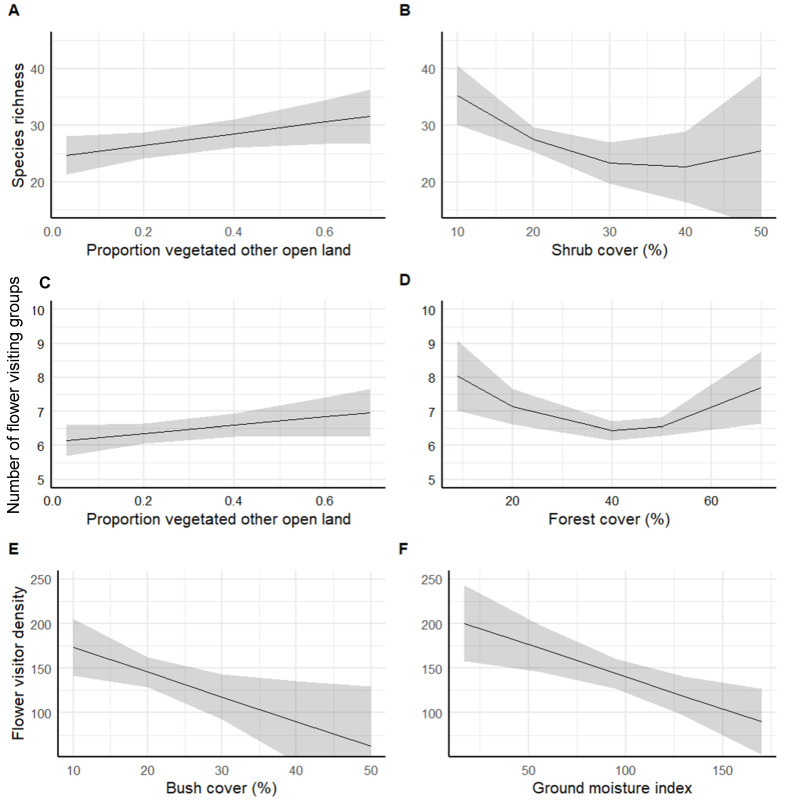
Species richness (**A**,**B**), flower-visiting group richness (**C**,**D**), and pollinator densities (**E**,**F**) with landscape variables from the best-fitting GLM. The model tests both linear and quadratic terms of landscape variables. The results from the statistical evaluation of the associations are reported in Table 1. Each subplot includes a line representing the model’s fitted values, with shaded areas denoting the 95% confidence intervals.

**Figure 4 insects-15-00563-f004:**
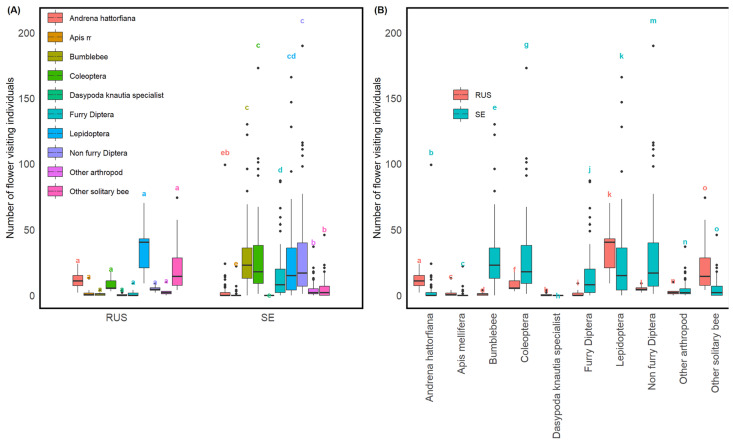
Number of individuals of ten different pollinator groups in Sweden and Russia. (**A**) Comparison of countries for each pollinator group. Each box represents a country, separated by pollinator group. (**B**) Comparison of pollinator groups between countries. Each box represents a pollinator group, separated by country. In the boxplots, the box represents the interquartile range (IQR) containing the middle 50% of the data, with the median shown as a horizontal line within the box. The lower and upper hinges of the box correspond to the first and third quartiles (25th and 75th percentiles). The whiskers extend to the largest and smallest values no further than 1.5 * IQR from the hinges. Points beyond the whiskers represent outliers. Lowercase letters indicate statistically significant differences between groups based on post hoc analysis. Groups sharing the same letter are not significantly different from each other, while groups with different letters are significantly different (*p* < 0.05). Statistical comparisons were performed using Welch’s *t*-test with Bonferroni correction. For results from post hoc analyses of respective groups, see Appendix A. RUS = Russia, SE = Sweden.

**Table 1 insects-15-00563-t001:** The best-fitting (lowest AIC) for the linear models testing species richness (R^2^ = 25%), morphogroup richness (R^2^ = 25%), and flower visitor desnity (R^2^ = 16%) against landscape variables.

Model	Predictor	Estimate	Std. Error	t Value	*p*-Value
Species richness	(Intercept)	25.231	1.721	14.658	<0.001
	Vegetated other open land	10.382	5.034	2.062	0.045
	Shrub cover	−18.076	6.567	−2.753	0.008
	Shrub cover^2^	10.633	6.51	1.633	0.10972
Morphogroup richness	(Intercept)	6.3602	0.2276	27.939	<0.001
	Vegetated other open land	1.1971	0.7567	1.582	0.117
	Forest cover	−0.8701	1.0885	−0.799	0.426
	Forest cover^2^	2.7907	0.9677	2.884	0.005
Flower visitor density	(Intercept)	272.9613	32.2213	8.471	<0.001
	Shrub cover	−2.7732	1.1654	−2.38	0.019
	Ground moisture index	−0.7205	0.2368	−3.042	0.003

## Data Availability

The data that support the findings of this study are available from the corresponding author upon request.

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
