# Peer review of "Exploring Biodiversity through the Lens of Knautia arvensis Pollinators: Knautia Pollinator Walks as a Monitoring Method"

_insects, 2024, doi:10.3390/insects15080563_

Round 1

Reviewer 1 Report

Comments and Suggestions for Authors

The paper presented to me is an interesting and valuable addition to pollinator research. I only have a few questions or corrections to the Authors.

Methods:

How long did the walks last?

Did you correct data for patch size, especially, when populations were smaller than 500 flower units? If not can you explain if it could have an effect on the sampling results?

Can you add the description of the predefined pollinator groups in the form of a table? 

You extracted landcover data for 106 sites but sampled only 77. Can you explain why did you drop out some of the sites?

Can you add a general description of where these populations were found? Were only semi-natural/natural environments or agricultural, urban surroundings also considered?

The Russian and the Swedish sites clearly differed. Why didn't you add geographic region as a predictor in the analysis?

Results:

line 226 "Within the 47 sites ...." rather 77? -

In 3.1, when describing the visitation rates and the diversity of visitors, do you mean during 3 visits in the season total or during one single visit?

Can you add the list of visitors on Knautia as supplementary material to the paper?

Discussion

line 289 Were there 47 or 77 Swedish sites? These two numbers appear, and I am not sure which is correct.

Author Response

Dear Referee,

Thank you for your valuable comments and suggestions, which have helped us improve our manuscript. Below, we address each of your points in detail.

Methods:

  1. Duration of Walks: We have added this information to the Methods section for clarity. See marked up copy.
  2. Correction for Patch Size: We did not correct the data specifically for patch size when populations were smaller than 500 flower units instead we repeated the walks until 500 500 flower units was reached. Very few population had fewer than 500 flower units.
  3. Description of Predefined Pollinator Groups: We have included a table (Table S1) in the revised manuscript with the identified flower visitors.
  4. We have clarified that 77 sites were sampled. We have added this explanation to the Methods section.
  5. General Description of Populations: We have added a description of the grasslands sampled. See the marked up copy for details.
  6. Geographic Region as a Predictor: We separated the regions and made post hoc tests. See figure 4. Its a matter of taste how to analyse region.  

Results:

  1. Line 226 Correction: We apologize for the confusion. The correct number should be 77 sites, not 47. We have corrected this in the manuscript.
  2. Visitation Rates and Diversity: In section 3.1, the visitation rates and diversity of visitors refer to the total observations made during the three visits in the season. We have clarified this point in the Results section.
  3. List of Visitors: We have added the list of visitors on Knautia as supplementary material to the paper. This list provides detailed information on the pollinator species observed during the study.

Discussion:

  1. Line 289 Correction: The correct number is 77 Swedish sites. We have corrected this discrepancy in the Discussion section.

We appreciate your careful review and constructive feedback, which have significantly enhanced the quality of our manuscript. We hope that the revisions meet your expectations and improve the clarity and impact of our study.

Sincerely,

Markus Franzén

Reviewer 2 Report

Comments and Suggestions for Authors

Declining pollinator number and diversity is the major concern of the pollination scientists. Recent studies reveal such declines in various parts of the globe. Their persistent monitoring is very important. The present article has attempted at this proposition. This is an excellent study very carefully planned and executed. I don’t find any lacunae in the research and its presentation in this article. Abstract and Introductions have been written very nicely. Materials and Methods are appropriate. Results have been presented lucidly in Tables and Figures. Discussion is to the point and relevant to the Results. Conclusion is appropriate and relevant references have been cited.

Author Response

Dear Referee,

Thank you for your positive feedback on our manuscript. We are delighted to hear that you found our study well-planned and executed. Thank you.

Sincerely,

Markus Franzen